# Molecular Detection and Characterization of *Blastocystis* sp. and *Enterocytozoon bieneusi* in Cattle in Northern Spain

**DOI:** 10.3390/vetsci8090191

**Published:** 2021-09-11

**Authors:** Nadia Abarca, Mónica Santín, Sheila Ortega, Jenny G. Maloney, Nadja S. George, Aleksey Molokin, Guillermo A. Cardona, Alejandro Dashti, Pamela C. Köster, Begoña Bailo, Marta Hernández-de-Mingo, Aly S. Muadica, Rafael Calero-Bernal, David Carmena, David González-Barrio

**Affiliations:** 1Parasitology Reference and Research Laboratory, National Centre for Microbiology, Majadahonda, 28220 Madrid, Spain; nadia.abarca@uacj.mx (N.A.); SHEILAORTEGA@isciii.es (S.O.); dashti.alejandro@gmail.com (A.D.); pamelakster@yahoo.com (P.C.K.); BEGOBB@isciii.es (B.B.); martaher1@hotmail.com (M.H.-d.-M.); muadica@gmail.com (A.S.M.); 2Department of Veterinary Sciences, Biomedical Sciences Institute, Autonomous University of Ciudad Juárez, Chihuahua 32310, Mexico; 3Environmental Microbial and Food Safety Laboratory, Agricultural Research Service, United States Department of Agriculture, Beltsville, MD 20705, USA; monica.santin-duran@usda.gov (M.S.); jenny.maloney@usda.gov (J.G.M.); nadja.george@usda.gov (N.S.G.); aleksey.molokin@usda.gov (A.M.); 4Livestock Laboratory, Regional Government of Álava, 01520 Vitoria-Gasteiz, Spain; gcardona@araba.eus; 5Departamento de Ciências e Tecnologia, Universidade Licungo, Beira-Quelimane Beira 2100, Zambézia, Mozambique; 6SALUVET, Department of Animal Health, Faculty of Veterinary, Complutense University of Madrid, 28040 Madrid, Spain; r.calero@ucm.es

**Keywords:** *Blastocystis*, *Enterocytozoon bieneusi*, cattle, NGS, ribosomal RNA, subtypes, Spain, zoonoses, transmission

## Abstract

Some enteric parasites causing zoonotic diseases in livestock have been poorly studied or even neglected. This is the case in stramenopile *Blastocystis* sp. and the microsporidia *Enterocytozoon bieneusi* in Spain. This transversal molecular epidemiological survey aims to estimate the prevalence and molecular diversity of *Blastocystis* sp. and *E. bieneusi* in cattle faecal samples (*n* = 336) in the province of Álava, Northern Spain. Initial detection of *Blastocystis* and *E. bieneusi* was carried out by polymerase chain reaction (PCR) and Sanger sequencing of the small subunit (*ssu*) rRNA gene and internal transcribed spacer (ITS) region, respectively. Intra-host *Blastocystis* subtype diversity was further investigated by next generation amplicon sequencing (NGS) of the *ssu* rRNA gene in those samples that tested positive by conventional PCR. Amplicons compatible with *Blastocystis* sp. and *E. bieneusi* were observed in 32.1% (108/336, 95% CI: 27.2–37.4%) and 0.6% (2/336, 95% CI: 0.0–1.4%) of the cattle faecal samples examined, respectively. Sanger sequencing produced ambiguous/unreadable sequence data for most of the *Blastocystis* isolates sequenced. NGS allowed the identification of 10 *Blastocystis* subtypes including ST1, ST3, ST5, ST10, ST14, ST21, ST23, ST24, ST25, and ST26. All *Blastocystis*-positive isolates involved mixed infections of 2–8 STs in a total of 31 different combinations. The two *E. bieneusi* sequences were confirmed as potentially zoonotic genotype BEB4. Our data demonstrate that *Blastocystis* mixed subtype infections are extremely frequent in cattle in the study area. NGS was particularly suited to discern underrepresented subtypes or mixed subtype infections that were undetectable or unreadable by Sanger sequencing. The presence of zoonotic *Blastocystis* ST1, ST3, and ST5, and *E. bieneusi* BEB4 suggest cross-species transmission and a potential risk of human infection/colonization.

## 1. Introduction

Some enteric parasites causing zoonotic diseases in livestock in Spain have been poorly studied or even neglected. This is the case in stramenopile *Blastocystis* sp. and the microsporidia *Enterocytozoon bieneusi*. *Blastocystis* sp. is probably the most common enteric parasite in humans, affecting more than 1 billion people globally [1]. Human *Blastocystis* infection has been associated with a variety of gastrointestinal symptoms, including diarrhoea, nausea, and abdominal pain [2]. However, the clinical and public health significance of *Blastocystis* sp. remains controversial as the protist is commonly found in both apparently healthy individuals and patients suffering from intestinal and extra-intestinal manifestations [3]. Members of the phylum microsporidia are obligate intracellular fungus-like parasites of worldwide distribution and have a marked high genetic diversity among mammalian and avian hosts. Of the 17 microsporidian species known to be infective to humans, zoonotic *E. bieneusi* is the most prevalent one, affecting the gastrointestinal tract and causing diarrhoea, mainly in immune-deficient individuals [4,5,6]. Among them, AIDS patients and children are especially susceptible to *E. bieneusi* infection [7].

Infections by *Blastocystis* sp. and *E. bieneusi* occur via the faecal-oral route through direct contact with infected hosts or due to consumption of food or water contaminated with cysts or spores [8,9,10], thus transmission through environmental contamination by competent animal host species is an important aspect of their epidemiology.

The genetic diversity and host specificity/range of both parasites have been the focus of intense research in recent years. At present, 27 genetically distinct small subunit ribosomal RNA (*ssu*) lineages or subtypes (STs) have been distinguished within *Blastocystis* sp. [11,12,13], and this number is likely to increase in the near future. Of them, ST1-ST4 account for ca. 90% of the human infections reported globally [14]. ST5 is commonly isolated from livestock, ST6 and ST7 from birds, and ST8 from arboreal non-human primates [15,16,17]). In addition to humans, ST9 has been recently described in poultry and non-human primates [18,19]. The fact that ST5‒ST8, ST10, ST12, and ST14 have been only sporadically found in humans has been interpreted as indicative of zoonotic activity [20,21,22]. ST11, ST13, and ST15‒ST17 have only been documented in non-human species so far [20,23]. Similarly, ST21 and ST23‒ST26 have been isolated from domestic and wild ruminants [11], ST27-ST29 in avian species [24,25], and ST30 and ST31 in wild ungulates [13].

On the other hand, DNA sequence analysis of the ribosomal internal transcribed spacer (ITS) region is widely used for the genetic characterization of *E. bieneusi* isolates and the assessment of infection sources, transmission, and zoonotic potential [7]. More than 500 genotypes of *E. bieneusi* have been described and allocated into 11 distinct genetic groups [4,26]. Groups 1 and 2 contain potentially zoonotic genotypes, whereas the remaining (Groups 3‒11) comprise host-adapted genotypes associated with specific animal species [4,7]. Humans are infected predominately by genetic variants within Group 1, which has over 300 genotypes, many of them with a loose host range. Group 2, in contrast, contains nearly 100 genotypes and was initially regarded as a ruminant-specific group [7].

In Spain, the molecular epidemiology of *Blastocystis* sp. and *E. bieneusi* is largely unknown. Few surveys have attempted to investigate the genetic diversity of these protist species in humans (including paediatric and clinical) [27,28,29]. Regarding animal hosts, *Blastocystis* sp. and *E. bieneusi* have been reported in livestock (pigs) [30], pets (dogs, cats) [31], and captive non-human primates [32] and animal populations and wildlife (leporids, carnivores) [33,34]. In contrast, only a single microscopy-based survey has investigated the presence of *Blastocystis* sp. in cattle [35], whereas no studies have ever attempted to identify the presence of *E. bieneusi* in this host in Spain. Although limited, previous studies conducted globally have demonstrated that cattle can harbour both zoonotic and enzootic *Blastocystis* subtypes and *E. bieneusi* genotypes [7,16,17,36,37,38,39,40,41]. The molecular characterization of *Blastocystis* sp. and *E. bieneusi* in Spanish cattle has not been investigated, nor has the potential role of this host as a natural reservoir of human blastocystosis or microsporidiosis by *E. bieneusi* been elucidated. Therefore, the aim of this cross-sectional epidemiological study was to determine the occurrence, genetic diversity, and zoonotic potential of *Blastocystis* sp. and *E. bieneusi* in cattle in the province of Álava, Northern Spain.

## 2. Materials and Methods

### 2.1. Sampling

Faecal samples from asymptomatic adult dairy and beef cattle were collected from the rectum using disposable plastic bags taking advantage of routine bovine paratuberculosis control programmes in 2019 (July) and 2020 (April and August) in Álava, Northern Spain. The investigated cattle belonged to the counties of Añana, Ayala, Agurain, and Zuia (Figure 1). Collected faecal samples were transported in cooled boxes to the Livestock Laboratory of the Regional Government of Álava in Vitoria-Gasteiz, Álava (Spain) for subsequent analysis. Samples were processed within 72 h of collection.

### 2.2. DNA Extraction and Purification

Total DNA was extracted from an aliquot (∼1 g) of each faecal sample using the DNA Extract-VK kit (Vacunek, Derio, Spain) according to the manufacturer’s instructions. Purified DNA samples (100 µL) were stored at −20 °C and shipped to the Parasitology Reference and Research Laboratory of the National Centre for Microbiology, Majadahonda (Spain) for further molecular analysis.

### 2.3. Molecular Detection of Blastocystis *sp.*

Identification of *Blastocystis* sp. was achieved by a direct PCR protocol targeting the *ssu* rRNA gene of the parasite [42]. The assay uses the pan-*Blastocystis*, barcode primer pair BhRDr (5’–GAGCTTTTTAACTGCAACAACG–3´) and RD5 (5´–ATCTGGTTGATCCTGCCAGT–3´) to amplify a PCR product of ca. 600 bp. Amplification reactions (25 μL) included 5 μL of template DNA and 0.5 μM of each primer. Cycling conditions consisted of one step of 95 °C for 3 min, followed by 30 cycles of 1 min each at 94, 59, and 72 °C, with an additional 2 min final extension at 72 °C.

### 2.4. Molecular Detection of Enterocytozoon bieneusi

Detection of *E. bieneusi* was conducted by a nested PCR protocol to amplify the internal transcribed spacer (ITS) region as well as portions of the flanking large and small subunit of the ribosomal RNA gene, as previously described [43]. The outer primer sets EBITS3 (5´–GGTCATAGGGATGAAGAG–3´) and EBTIS4 (5´–TTCGAGTTCTTTCGCGCTC–3´) and the inner primer sets EBITS1 (5´–GCTCTGAATATCTATGGCT–3´) and EBITS2.4 (5´–ATCGCCGACGGATCCAAGTG–3´) were used to generate a final PCR product of ca. 390 bp. Cycling conditions for the primary PCR consisted of one step of 94 °C for 3 min, followed by 35 cycles of amplification (denaturation at 94 °C for 30 s, annealing at 57 °C for 30 s, and elongation at 72 °C for 40 s), with a final extension at 72 °C for 10 min. Conditions for the secondary PCR were identical to the primary PCR, except that only 30 cycles were carried out with an annealing temperature of 55 °C.

All the PCR protocols described above were conducted on a 2720 Thermal Cycler (Applied Biosystems, Foster City, CA, USA). Reaction mixes always included 2.5 units of MyTAQ^TM^ DNA polymerase (Bioline GmbH, Luckenwalde, Germany) and 5× MyTAQ^TM^ Reaction Buffer containing 5 mM dNTPs and 15 mM MgCl_2_. Laboratory-confirmed positive and negative DNA samples for each parasitic species investigated were routinely used as controls and included in each round of PCR. PCR amplicons were visualized on 2% D5 agarose gels (Conda, Madrid, Spain) stained with Pronasafe nucleic acid staining solution (Conda). A 100 bp DNA ladder (Boehringer Mannheim GmbH, Baden-Wurttemberg, Germany) was used for the sizing of the obtained amplicons.

### 2.5. Sanger Sequencing and Sequence Analysis

Positive-PCR products of the expected sizes were sequenced in both directions using internal primer sets. Capillary DNA sequencing electrophoresis were used, utilizing BigDye^®^ Terminator chemistry on an ABI PRISM 3130 automated DNA sequencer (Applied Biosystems, Foster City, CA, USA).

Raw sequencing data in both forward and reverse directions were viewed using the Chromas Lite version 2.1 (TechnelysiumPty Ltd., South Brisbane, Australia) sequence analysis program (https://technelysium.com.au/wp/chromas/, accessed on 10 September 2021). The BLAST tool (http://blast.ncbi.nlm.nih.gov/Blast.cgi, accessed on 10 September 2021) was used to compare nucleotide sequences with those retrieved from the NCBI GenBank database. Generated DNA consensus sequences were aligned to appropriate reference sequences using the MEGA version 6 software [44] to identify *Blastocystis* STs and *E. bieneusi* genotypes. The *E. bieneusi* sequences obtained in this study have been deposited in GenBank under accession number MZ666878.

### 2.6. Blastocystis Subtype Identification Using Next Generation Amplicon Sequencing

DNA aliquots of potential *Blastocystis*-positive samples by conventional *ssu*-PCR were shipped to the Environmental Microbial and Food Safety Laboratory, Agricultural Research Service, Beltsville, MD, (USA). A next generation amplicon sequencing strategy was used to assess intra-isolate *Blastocystis* molecular diversity as previously described [45]. Briefly, a PCR using primers ILMN_Blast505_532F (5′*–*TCGTCGGCAGCGTCAGATGTGTATAAGAGACAGGGAGGTAGTGACAATAAATC*–*3′) and ILMN_Blast998_1017R (5′*–*GTCTCGTGGGCTCGGAGATGTGTATAAGAGACAGTGCTTTCGCACTTGTTCATC*–*3′) was used to screen all positive/suspected samples by conventional PCR. These primers amplify a fragment of the *ssu* rRNA gene (ca. 500 bp) and are identical to Blast505_532F/Blast998_1017R [46], with the exception that they contain the Illumina overhang adapter sequences (underlined) on the 5′ end. Final libraries were quantified by A Qubit (Invitrogen, Carlsbad, CA, USA) was used to fluorometrically quantify the sequencing libraries in order to normalize their concentration. Libraries were then pooled and diluted again to an 8 pM final loading concentration and spiked with 20% PhiX control. Sequencing was performed using an Illumina MiSeq using 600 cycle, v3 chemistry (Illumina, San Diego, CA, USA). Read pairs generated by the MiSeq were processed using an in-house pipeline that leveraged the following open source linux tools: BBTools package v38.82 [47], VSEARCH v2.15.1 [48], and BLAST+ 2.11.0 [49]. The processing steps included the merging of read pairs, quality and length filtering, sequence denoising, and chimera filtering. Filtered reads were then clustered into operational taxonomic units (OTUs) within each sample using a 98% identity threshold. Any OTU with an abundance below 100 sequences was filtered out and the remaining OTUs were once more checked for chimeric sequences within each sample. OTUs were then aligned to Blastocystis references from NCBI using Blast+. An alignment length cut-off of 400 bp was used to filter out short blast hits. Raw FASTQ files were submitted to NCBI’s sequence read archive under project PRJNA747253 and accession numbers SRR19164320–SRR19164427. Final OTU sequences generated by this study were submitted to GenBank under the accession numbers MZ664502–MZ664552.

### 2.7. Statistical Analysis

The prevalence of *Blastocystis* sp. and *E. bieneusi* infections, including 95% confidence intervals (95% CI), was calculated. The chi-squared test was used to compare *Blastocystis* sp. and *E. bieneusi* infection rates by the production system (dairy or beef) or county of origin of the surveyed cattle population. A *p* value < 0.05 was considered evidence of statistical significance. Analyses were conducted on R software version 3.6.0 (R Foundation for Statistical Computing, Austria, Vienna).

## 3. Results

### 3.1. Detection of Blastocystis sp. and E. bieneusi by PCR

A total of 336 faecal samples from dairy (50.0%, 168/336) and beef (19.3%, 65/336) cattle were collected during the study period. The remaining faecal samples (30.7%, 103/336) were from cattle whose production system was unknown. According to the county of origin, sampled animals were from Añana (3.3%, 11/336), Ayala (31.5%, 106/336), Llanada Alavesa (0.3%, 1/336), and Zuia-Gorbialdea (35.1%, 118/336). Origin was unknown in 29.8% (100/336) of the surveyed cattle due to incomplete sample registration at sampling (Table 1).

Overall, 32.1% (108/336, 95% CI: 27.2–37.4%) of the bovine samples yielded amplicons of the expected size by *ssu*-PCR using the pan-*Blastocystis* primers BhRDr/RD5. No statistically significant differences in *Blastocystis* carriage/infection rates were observed between dairy and beef cattle (*p* = 0.127). However, cattle from the Añana county harboured significantly higher *Blastocystis* carriage/infection rates than their counterparts from other counties (*p* = 0.0226). *Enterocytozoon bieneusi* was identified in 0.6% (2/336, 95% CI: 0.0–1.4%) of the samples analysed (Table 1), but its distribution according to production system or county of origin was not statistically assessed due to the low number of positive samples and lack of statistical power.

### 3.2. Molecular Characterization of Blastocystis sp. and E. bieneusi by Sanger Sequencing

Out of the 108 bovine samples that yielded potential *Blastocystis*-positive results by *ssu*-PCR, only 34.3% (37/108) could be confirmed by Sanger sequencing. Sequence analyses revealed ST10 as the most prevalent *Blastocystis* ST circulating in the bovine population under investigation (51.4%, 19/37), followed by ST5 (45.9%, 17/37) and ST14 (2.7%, 1/37). Near half (44.7%, 17/38) of the subtyped samples belonged to the cattle group for which management production was unknown (Table 2). Allele calling using the *Blastocystis* 18S online database failed to clearly determine intra-subtype allelic diversity due to the presence of ambiguous (double peak) positions. The remaining 71 isolates corresponded to poor quality sequences associated to faint bands on agarose gels (that is, a sub-optimal amount of parasitic DNA), or unreadable sequences associated with the presence of mixed sequence traces, likely representing mixed infections.

Nucleotide sequence analysis of the two samples that tested positive for *E. bieneusi* by ITS-PCR revealed the presence of the BEB4 genotype, a genetic variant belonging to the zoonotic Group 2 of the parasite. These two samples were identified in dairy cattle (Table 2).

### 3.3. Blastocystis Subtype Identification by Next Generation Amplicon Sequencing

All *Blastocystis*-positive samples (*n* = 108) confirmed by Sanger sequencing (*n* = 37) or *Blastocystis*-suspected (*n* = 71) samples by *ssu*-PCR (unconfirmed by Sanger sequencing due to poor quality of chromatograms) were subjected to NGS analysis to further confirm the presence of the parasite and/or to determine the diversity and frequency of subtypes. NGS analysis confirmed the presence of *Blastocystis* sp. in 108 samples and allowed the identification of 10 distinct *Blastocystis* STs including ST1, ST3, ST5, ST10, ST14, ST21, and ST23–ST26 circulating in the bovine population under investigation (Table 3). These represent seven STs more than those detected by Sanger sequencing (Table 3). The three most common *Blastocystis* STs observed by NGS were ST10 (98.2%, 106/108), ST25 (86.1%, 93/108), and ST26 (92.6%, 101/108). Potentially zoonotic subtypes ST1 (1.9%, 2/108), ST3 (0.9%, 1/108), and ST5 (13.0%, 14/108) were observed in the surveyed bovine population; among them, only ST5 was also detected by Sanger sequencing.

Remarkably, NGS analyses revealed that all *Blastocystis*-positive cattle carried mixed infections involving 2–8 STs of the parasite in 31 different combinations (Table 4 and Appendix A). The most common mixed infections were those involving ST10 + ST25 + ST26 (13.9%, 15/108), ST10 + ST14 + ST21 + ST25 + ST26 (13.9%, 15/108), and ST10 + ST14 + ST21 + ST23 + ST25 + ST26 (11.1%, 12/108).

## 4. Discussion

The epidemiology of the stramenopile *Blastocystis* sp. and the microsporidia *E. bieneusi* in livestock in general, and cattle in particular, is largely unexplored in Spain. In the case of cattle, previous studies only used conventional microscopy for detection, so the genetic variants of these enteric parasites circulating in bovines remains unknown. This molecular-based epidemiological study describes, for the first time in the country, the genetic diversity of *Blastocystis* sp. and *E. bieneusi* in cattle in Northern Spain.

In Spain, few studies have examined by conventional (formaline-ethyl acetate sedimentation or Faust flotation concentration and light microscopy) methods the presence of *Blastocystis* sp. in livestock and captive animals. These surveys were constricted by limited diagnostic sensitivity, so documented *Blastocystis* occurrence rates were likely underestimated. Thus, prevalence rates of 1.8% (10/554) were reported in cattle from 30 farms in Aragon (Northeaster Spain) [35], and of 7.5% (27/360) in swine from 17 farms in the same region [50]. *Blastocystis* sp. has also been described at a prevalence rate of 58.0% (291/502) in ostriches from 111 Spanish farms [51]. In addition, *Blastocystis* sp. has been found in 15.4% (67/432) of captive animals in a zoo in Southern Spain [52], in 23.6% (232/984) of birds in an ornithological garden in the same region [53], and in non-human primates (NHP) at the Barcelona Zoo at an unspecified carriage rate [54]. A higher prevalence rate of 45.1% (23/51) was reported by PCR and Sanger sequencing methods in NHP from the Córdoba Zoo Conservation Centre in Southern Spain [32].

The relatively high *Blastocystis* prevalence rate (32.1%) found in adult, apparently healthy cattle in the present study agrees with previous surveys suggesting that there is a positive correlation between the age of the animal and the prevalence of the parasite [41,55]. In the case of dairy cattle, this observation could potentially be due to the lower exposure of calves to the parasite, as they are fed under more hygienic conditions than heifers or adult animals. Our prevalence data is also in line with those reported globally in cattle. Indeed, a recent systematic review and meta-analysis, in which 28 papers were considered, an average *Blastocystis* prevalence of 24.4% was estimated in this host species [56].

Based on the genetic variability within the *ssu* rRNA gene, at least 27 distinct *Blastocystis* STs have been identified to date [11,12,13]. Recently, third-generation sequencing technologies such as the Oxford Nanopore MinION have been used for obtaining of the full-length sequences of the *ssu* rRNA gene required to unambiguously identify and characterize novel *Blastocystis* STs from difficult matrices such as faecal material [57]. In Spain, molecular characterisation studies of *Blastocystis* are still limited. Regarding animal hosts, *Blastocystis* STs have been found in faecal samples from Iberian pigs (ST1, ST3, and ST5) and wild boars (ST5) living sympatrically [30]. A high genetic variability including ST1, ST2, ST4, ST7, and ST14 was observed in different wild carnivore species in different regions of the country [58], whereas ST1, ST3, and ST8 were reported as circulating in captive NHP [32]. Most *Blastocystis* STs described in Spanish animal populations were potentially zoonotic, particularly ST1–ST5. In Spain, companion dogs and cats do not seem to play a significant role as suitable hosts for *Blastocystis* sp. [28].

In our study, we were able to identify ten different *Blastocystis* STs by NGS, and six of them (ST10, ST21, ST23–ST26) have not been described before in Spain. This is in sharp contrast to our results using Sanger sequencing, where only three STs (ST5, ST10, and ST14) were detected and a large proportion (65.7%, 71/108) of the samples suspected to be *Blastocystis*-positive generated unreadable chromatograms due to overlapping sequences. Taken together, these findings indicate that (i) mixed infections involving different *Blastocystis* STs are extremely common in cattle, (ii) NGS is able not only to detect underrepresented STs in the population under study, but also to effectively discriminate among the STs involved in mixed infections, and (iii) Sanger sequencing is only effective for ST determination if a single ST is the dominant amplicon in a sample.

A wide *Blastocystis* genetic diversity has been documented in cattle worldwide. Sixteen different subtypes (ST1–ST7, ST10, ST12, ST14, ST17, ST21, ST23–ST26) have been de-scribed (at different frequency rates) from cattle worldwide, indicating that cattle can act as a suitable reservoir for many *Blastocystis* genetic variants, including several with zoonotic potential [56]. Among the subtypes reported in cattle, ST10 has been demonstrated as the most prevalent subtype in this host [59]. This is also the case for the present study, where ST10 was present in nearly all (98.2%) of the cattle analysed. Other frequent STs included ST26 (93.5%), ST25 (86.1%), ST21 (64.8%), and ST14 (63.0%). Of note, both ST25 and ST26 were detected in pre-weaned dairy heifer calves in the USA at much lower frequency rates [41], suggesting that the frequency of these STs may vary geographically or by host age. Remarkably, zoonotic ST1, ST3, and ST5 were also present in the bovine population under study, indicating that several sources of infection or even transmission pathways may be involved in the epidemiology of the parasite in this Spanish region. Studies from Iran and the USA also reported ST5 as a common finding in cattle [16,41]. The zoonotic potential of this *Blastocystis* ST was demonstrated in surveys conducted in Australia and China, where ST5 was not only found in swine but also in their caretakers [60,61,62].

*Enterocytozoon bieneusi* has been described in Spain in a wide range of animal hosts including companion (cats and dogs), production (goat, swine, rabbits, and ostriches) and free-living (pigeons, foxes) animal species [31,34,63,64,65,66], as well as in immunocompromised [27,67,68] and immunocompetent [69,70,71] individuals. In addition, spores of *E. bieneusi* have been identified in environmental (water) samples [72]. In the present survey, *E. bieneusi* was identified in 0.6% of the investigated cattle. This constitutes the first report of *E. bieneusi* in this host species in Spain. Worldwide, there is an abundant literature on the detection of *E. bieneusi* in cattle, where the reported prevalence in European countries typically varied in the range of 6–15%, which is considerably higher than that reported here [7]. An overall microsporidia (mostly attributable to *E. bieneusi*) prevalence in cattle of 16.6% (2,216/12,175) has been recently estimated in a large meta-analysis study including data from 79 published studies from 17 countries [73]. Studies conducted in China did not find significant differences in the occurrence of *E. bieneusi* among beef cattle, dairy cattle, and water buffaloes [37,39], although the opposite was true in a survey conducted in Brazil, where dairy cattle (26.2%) had a higher prevalence than beef cattle (9.7%) [38]. Domestic ruminants show a diverse pattern of genotype distribution; they are infected predominantly by Group 2 genotypes, especially BEB4 (the same *E. bieneusi* genotype detected in the present study), BEB6, I, and J [7]. Thus, BEB4 has been identified as infecting cattle in Argentina [74], Brazil [38], China [75], Korea [76], South Africa [77], and the USA [78]. In addition to cattle, genotype BEB4 has also been reported in pigs, yaks, donkeys, and NHP [75,79,80,81]. Furthermore, two surveys have documented the presence of BEB4 circulating in humans from China and the Czech Republic, raising concerns regarding its zoonotic potential [75,82]. It is noteworthy that domestic ruminants can also harbour well-known zoonotic Group 1 genotypes, such as EbpC, D and Type IV [83,84]; therefore, it has been indicated that *E. bieneusi* infections in persons from Zimbabwe may result from exposure to cattle faeces [85], suggesting a potential role of ruminants in the zoonotic transmission of *E. bieneusi*. In our study, we have identified only two positive samples whose genotype belongs to BEB4, within Group 2. This finding may suggest a role for cattle as a potential source of human infections by *E. bieneusi*, although the extent of this possibility must be confirmed in future molecular epidemiological surveys tackling bovine and human populations sharing the same environment.

## 5. Conclusions

This is the first molecular-based study specifically aimed at investigating the frequency, diversity, and zoonotic potential of *Blastocystis* subtypes and *E. bieneusi* genotypes circulating in Spanish cattle. We have demonstrated that *Blastocystis* infections involving up to eight different subtypes are extremely common in cattle, including some with well-recognized zoonotic potential such as ST1, ST3, and ST5. Importantly, we have corroborated previous molecular epidemiological studies demonstrating the usefulness and practicality of NGS for detecting underrepresented STs or complex mixed infections that would be overlooked by Sanger sequencing. *Enterocytozoon bieneusi* was detected at low infection rates by the zoonotic BEB4 genotype of the parasite. Overall, the molecular data presented here describes an epidemiological scenario in which different sources of infection and transmission pathways are taking part in the epidemiology of *Blastocystis* sp. and *E. bieneusi*. Given the relatively limited sample size and restricted geographical location of the study, more research should be conducted in other Spanish cattle populations to corroborate and expand the findings presented here. Our results highlight the idea that cattle could play a significant role in the transmission of these parasites through contamination of ground and surface water with *Blastocystis* cysts and *E. bieneusi* spores and serve as suitable reservoirs for human and domestic animal infections.

## Figures and Tables

**Figure 1 vetsci-08-00191-f001:**
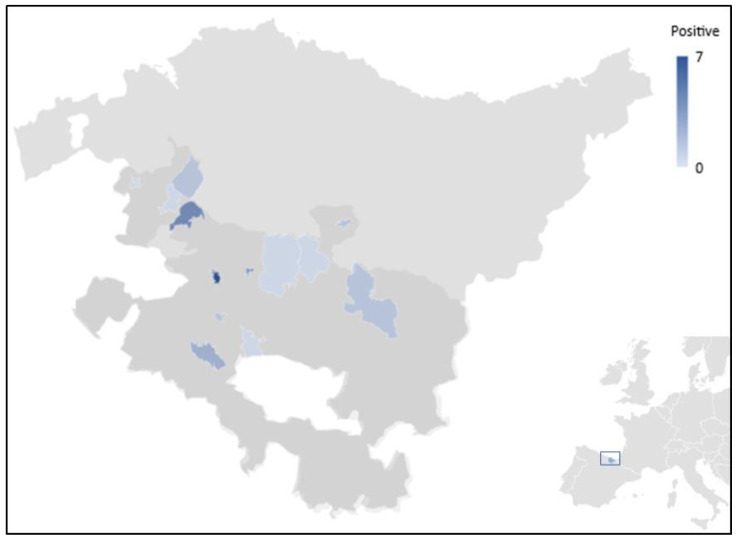
Map showing the geographical location of the sampling points included in the present study and the corresponding gradient of *Blastocystis*-positive results. The province of Álava (shaded in dark grey) is the Southernmost province of the Basque Autonomous Community in Northern Spain (highlighted in blue at the bottom right corner of the picture).

**Table 1 vetsci-08-00191-t001:** PCR-based infection rates of *Blastocystis* sp. and *Enterocytozoon bieneusi* in cattle according to production system, Álava (Spain) 2019–2020.

		*Blastocystis* sp.		*Enterocytozoon bieneusi*	
Variable	*n*	Positive (*n*)	%	Positive (*n*)	%
Production system					
Dairy	168	38	22.6	2	1.2
Beef	65	21	32.3	0	0.0
Unknown	103	49	47.6	0	0.0
County					
Añana	11	6	54.5	0	0.0
Ayala	106	20	18.9	0	0.0
Llanada Alavesa	1	1	100	0	0.0
Zuia-Gorbeialdea	118	32	27.1	2	1.7
Unknown	100	49	49.0	0	0.0
Total	336	108	32.1	2	0.6

**Table 2 vetsci-08-00191-t002:** Diversity and frequency of *Blastocystis* subtypes and *Enterocytozoon bieneusi* genotypes confirmed by Sanger sequencing in cattle according to production system, Álava (Spain) 2019–2020. Potentially zoonotic genetic variants are in bold.

	*Blastocystis* sp.		*Enterocytozoon bieneusi*	
Variable	Isolates(*n*)	Subtypes (*n*)	Isolates(*n*)	Genotypes (*n*)
Production system				
Dairy	11	**ST5** (4), ST10 (7)	2	**BEB4** (2)
Beef	9	**ST5** (4), ST10 (5)	0	–
Unknown	17	**ST5** (9), ST10 (7), ST14 (1)	0	–
Total	37		2	

**Table 3 vetsci-08-00191-t003:** Diversity and frequency of *Blastocystis* subtypes identified by Next Generation Amplicon Sequencing in cattle, Álava (Spain) 2019–2020. Results obtained by Sanger sequencing are shown for comparative purposes. Potentially zoonotic genetic variants are in bold.

Subtype	Next Generation Amplicon Sequencing	Sanger Sequencing
**ST1**	1.9% (2/108)	Not detected
**ST3**	0.9% (1/108)	Not detected
**ST5**	13.0% (14/108)	45.9% (17/37)
ST10 ^1^	98.2% (106/108)	51.4% (19/37)
ST14 ^1^	63.0% (68/108)	2.7% (1/37)
ST21	64.8% (70/108)	Not detected
ST23	35.2% (38/108)	Not detected
ST24	25.9% (28/108)	Not detected
ST25	86.1% (93/108)	Not detected
ST26	93.5% (101/108)	Not detected

^1^ Subtypes with zoonotic potential but rarely seen in humans. See reference [22].

**Table 4 vetsci-08-00191-t004:** *Blastocystis* subtype combinations observed by NGS in cattle, Álava (Spain) 2019–2020. Potentially zoonotic genetic variants are in bold.

Number of Subtypes in Co-Infection	Subtype Combinations	Samples (*n*)
2	ST10 + ST25	1
	ST25 + ST26	1
3	**ST5** + ST10 + ST25	1
	**ST5** + ST10 + ST14	1
	ST10 + ST14 + ST24	2
	ST10 + ST21 + ST26	1
	ST10 + ST25 + ST26	15
	ST14 + ST25 + ST26	1
4	**ST5** + ST10 + ST25 + ST26	3
	ST10 + ST14 + ST21 + ST24	1
	ST10 + ST14 + ST25 + ST26	5
	ST10 + ST21 + ST23 + ST26	1
	ST10 + ST21 + ST24 + ST26	1
	ST10 + ST21 + ST25 + ST26	4
	ST10 + ST23 + ST25 + ST26	4
5	**ST5** + ST10 + ST14 + ST25 + ST26	1
	**ST5** + ST10 + ST14 + ST24 + ST26	1
	ST10 + ST14 + ST21 + ST24 + ST26	4
	ST10 + ST14 + ST21 + ST25 + ST26	15
	ST10 + ST21 + ST23 + ST25 + ST26	8
	ST10 + ST14 + ST24 + ST25 + ST26	1
6	**ST5** + ST10 + ST14 + ST21 + ST25 + ST26	4
	**ST5** + ST10 + ST14 + ST23 + ST25 + ST26	1
	ST10 + ST14 + ST21 + ST23 + ST24 + ST25	1
	ST10 + ST14 + ST21 + ST23 + ST24 + ST26	3
	ST10 + ST14 + ST21 + ST23 + ST25 + ST26	12
	ST10 + ST14 + ST21 + ST24 + ST25 + ST26	5
	ST10 + ST21 + ST23 + ST24 + ST25 + ST26	1
7	**ST1** + **ST5** + ST10 + ST14 + ST21 + ST25 + ST26	1
	ST10 + ST14 + ST21 + ST23 + ST24 + ST25 + ST26	7
8	**ST3** + ST10 + ST14 + ST21 + ST23 + ST24 + ST25 + ST26	1

## Data Availability

All relevant data are within the article and its Appendix A.

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
