# Peer review of "Molecular Detection and Characterization of Blastocystis sp. and Enterocytozoon bieneusi in Cattle in Northern Spain"

_vetsci, 2021, doi:10.3390/vetsci8090191_

Round 1

Reviewer 1 Report

The article by  Abarca et al. determine the prevalence and genetic diversity of Blastocystis sp and Enterocytozoon bieneusi  in cattle in Northern Spain. The manuscript reads well and the title is appropriate.

I have the following comments:

  • Line 218-233: The manuscript would gain so much in value if a phylogenetic analysis of the gene sequences among STs would have been done. That would really be an added value of the manuscript. Therefore, I strongly recommend the authors to perform the phylogenetic analysis of the Sanger sequences obtained in the study together with those registered in GenBank in order to clearly see the different STs/clades and BEB4.
  • Line 238-239: “All Blastocytis-confirmed…. samples by ssu-PCR” is not correct, confirmation is done by microscopic examination and/or by DNA sequencing. Therefore change “ssu-PCR” to “DNA sequencing”.
  • Line 239: were subjected to NGS analysis to “further” confirm. Add “further” is the statement.
  • Table 4: Include “Location/County” where samples with each Subtype was found.

Also, is there any difference in the STs found among confirmed Blastocystis and suspected Blastocystis samples. The distinction should be made in the table 4.

Line 43: Keywords: limit to 5. Remove “NGS, Ribosomal RNA, Zoonoses, Transmission”

Author Response

Reviewer #1

The article by Abarca et al. determine the prevalence and genetic diversity of Blastocystis sp and Enterocytozoon bieneusi in cattle in Northern Spain. The manuscript reads well and the title is appropriate. I have the following comments:

We thank Reviewer #1 for his/her preliminary positive appraisal. We have carefully considered all the comments and suggestions made. Changes introduced in the text have been highlighted in red for better identification.

  1. Line 218-233: The manuscript would gain so much in value if a phylogenetic analysis of the gene sequences among STs would have been done. That would really be an added value of the manuscript. Therefore, I strongly recommend the authors to perform the phylogenetic analysis of the Sanger sequences obtained in the study together with those registered in GenBank in order to clearly see the different STs/clades and BEB4.

Response: Phylogenetic trees are useful to assess evolutionary relationships when a novel genetic variant of a given organism is discovered. In those circumstances, the clustering of the novel genetic variants with previously known sequences and the pattern of branching provides information about how these genetic variants evolved from a series of common ancestors. However, all Blastocystis STs (ST1, ST3, ST5, ST10, ST14, ST21, ST23, ST24, ST25, and ST26) and the E. bieneusi genotype (BEB4) found in the present study have been previously described in cattle molecular studies conducted in other geographical areas and do not constitute novel genetic variants.

  1. Line 238-239: “All Blastocytis-confirmed…. samples by ssu-PCR” is not correct, confirmation is done by microscopic examination and/or by DNA sequencing. Therefore change “ssu-PCR” to “DNA sequencing”.

Response: to improve clarity, the paragraph has been reworded as “All Blastocystis-positive samples (n = 108) confirmed by Sanger sequencing (n = 37) or Blastocystis-suspected (n = 71) samples by ssu-PCR (unconfirmed by Sanger sequencing due to poor quality of chomatograms) were subjected to…”.

  1. Line 239: were subjected to NGS analysis to “further” confirm. Add “further” is the statement.

Response: corrected as per requested by Reviewer #1.

  1. Table 4: Include “Location/County” where samples with each Subtype was found.

Response: please note Table 4 shows the 31 different combinations of the Blastocystis subtypes observed among the 77 Blastocystis isolates for which NGS data were available. In our opinion, showing in this very same Table the distribution of STs by county of origin will distract the reader as there are too many different combinations. Please note that this information is already provided in Table S1 in current lines 255 and 256.

  1. Also, is there any difference in the STs found among confirmed Blastocystis and suspected Blastocystis The distinction should be made in the Table 4.

Response: In Table 4 we are simply reporting all Blastocystis subtype combinations observed by NGS in all Blastocystis positives (n=108). It is not a sample-by-sample comparison of Sanger and NGS results. In Table S1 we have included the full dataset that includes details of PCR and sequencing results for Blastocystis sp. We have attempted to clarify this point in current line 166, lines 238-240, and lines 255-256 of the text.

  1. Line 43: Keywords: limit to 5. Remove “NGS, Ribosomal RNA, Zoonoses, Transmission”

Response: please note that, following the Intructions for Authors of Veterinary Science, “Keywords: Three to ten pertinent keywords need to be added after the abstract” (see https://www.mdpi.com/journal/vetsci/instructions). We have now 9 keywords, well in the range of the allowed number.

Reviewer 2 Report

The manuscript refers an interesting molecular survey on the epidemiology of Blastocystis and Enterocytozoon bieneusi in cattle from Northern regions of Spain. Data are very novel and interesting and some minor modifications are required.

The Authors should avoid to repeat taxonomic position of organisms investigated, this is redundant, so please modify throughout the whole text and, in particular (see below)

line 88 - the agents are not protists, please delete the word

line 212 - please delete "the microsporidia"

Author Response

Reviewer #2

The manuscript refers an interesting molecular survey on the epidemiology of Blastocystis and Enterocytozoon bieneusi in cattle from Northern regions of Spain. Data are very novel and interesting and some minor modifications are required.

We thank Reviewer #2 for his/her preliminary positive appraisal. We have carefully considered all the comments and suggestions made. Changes introduced in the text have been highlighted in red for better identification.

The Authors should avoid to repeat taxonomic position of organisms investigated, this is redundant, so please modify throughout the whole text and, in particular (see below).

  1. line 88 - the agents are not protists, please delete the word.

Response: Enterocytozoon bieneusi is a protist of the phylum Microsporidia and Blastocystis sp. is a protist belonging to the Stramenopiles. In our opinion, the use of the term “protist” through the whole manuscript is correct.

  1. line 212 - please delete "the microsporidia"

Response: removed as per requested by Reviewer #2.